# In Service Performance of Toughened PHBV/TPU Blends Obtained by Reactive Extrusion for Injected Parts

**DOI:** 10.3390/polym14122337

**Published:** 2022-06-09

**Authors:** Kerly Samaniego-Aguilar, Estefanía Sánchez-Safont, Alex Arrillaga, Jon Anakabe, Jose Gamez-Perez, Luis Cabedo

**Affiliations:** 1Polymers and Advanced Materials Group (PIMA), Universitat Jaume I, Av. Sos Baynat s/n, 12071 Castelló, Spain; samanieg@uji.es (K.S.-A.); esafont@esid.uji.es (E.S.-S.); jose.gamez@esid.uji.es (J.G.-P.); 2Leartiker S. Coop., Xemein Etorbidea 12, 48270 Markina-Xemein, Spain; aarrillaga@leartiker.com (A.A.); janakabe@leartiker.com (J.A.)

**Keywords:** PHBV, TPU, HMDI, toughening, rheology, biodegradability, biopolymer, immiscible blends

## Abstract

Moving toward a more sustainable production model based on a circular economy, biopolymers are considered as one of the most promising alternatives to reduce the dependence on oil-based plastics. Polyhydroxybutyrate-co-valerate (PHBV), a bacterial biopolyester from the polyhydroxialkanoates (PHAs) family, seems to be an attractive candidate to replace commodities in many applications such as rigid packaging, among others, due to its excellent overall physicochemical and mechanical properties. However, it presents a relatively poor thermal stability, low toughness and ductility, thus limiting its applicability with respect to other polymers such as polypropylene (PP). To improve the performance of PHBV, reactive blending with an elastomer seems to be a proper cost-effective strategy that would lead to increased ductility and toughness by rubber toughening mechanisms. Hence, the objective of this work was the development and characterization of toughness-improved blends of PHBV with thermoplastic polyurethane (TPU) using hexamethylene diisocyanate (HMDI) as a reactive extrusion agent. To better understand the role of the elastomer and the compatibilizer, the morphological, rheological, thermal, and mechanical behavior of the blends were investigated. To explore the in-service performance of the blends, mechanical and long-term creep characterization were conducted at three different temperatures (−20, 23, 50 °C). Furthermore, the biodegradability in composting conditions has also been tested. The results showed that HMDI proved its efficiency as a compatibilizer in this system, reducing the average particle size of the TPU disperse phase and enhancing the adhesion between the PHBV matrix and TPU elastomer. Although the sole incorporation of the TPU leads to slight improvements in toughness, the compatibilizer plays a key role in improving the overall performance of the blends, leading to a clear improvement in toughness and long-term behavior.

## 1. Introduction

Plastics are present in almost all areas of our lives. We can find them in applications ranging from automotive, construction, or electronics to biomedical or food packaging applications. Nowadays, the plastics market is clearly dominated by petroleum-based plastics with a current world annual production of 368 million tons for 2020 [1]. The extensive use of oil-based non-biodegradable plastics entails a series of environmental and socio-economic risks derived from both their non-renewable origin and high carbon footprint as well as the high volume of waste generated. In this sense, there is an urgent need to seek more sustainable alternatives to these products, either through the development of recyclable products (by traditional or organic recycling) or through the search for new renewable and/or biodegradable raw materials. Both approaches aimed at reducing their environmental impact from their origin to their end-of-life in a context based on the transition toward a more sustainable model inside a circular economy paradigm [2]. Within this context, biopolymers, defined as either bio-based or biodegradable polymers, are called to become the materials of the future to substitute commodities in a wide range of applications [3,4,5].

Polyhydroxyalkanoates (PHAs) are an interesting family of biopolymers that has attracted much attention in recent decades due to their natural origin, biodegradability, biocompatibility, and comparable physicochemical, thermal, and mechanical properties to some commodity plastics such as polyolefins [6]. PHAs can be synthesized by a wide range of microorganisms as carbon and energy reserve materials in the form of intracellular granules under metabolic stress conditions in the presence of a carbon source in excess [7,8]. Due to their bacterial origin, they can be metabolized by a multitude of microorganisms, showing high biodegradation rates in all environments including soil and marine [9,10]. Polyhydroybutyrate (PHB) and low hydroxyvalerate content polyhydroxybutyrate-co-hydroxyvalerate (PHBV) are the most widely studied PHAs. They are semi-crystalline thermoplastic polymers that can be processed by conventional plastic processing techniques such as extrusion, injection molding, thermoforming, film blowing, etc., with a high heat deflection temperature and balanced mechanical performance in terms of stiffness and tensile strength, similar to polypropylene (PP) [6,11]. Despite their obvious strengths and the numerous advances achieved through extensive research on these polymers, they still present some drawbacks that limit their industrial use. These are mainly related to a high production cost [12], a poor thermal stability with a narrow processing window, and low toughness and ductility [13,14,15,16], the latter being of particular importance from a technical point of view.

Regarding the cost reduction, different strategies can be found in the literature, either aimed at optimizing the production methods or aimed at combining PHAs with other cheaper raw materials to lower the price of the final product. In the first case, the use of mixed microbial cultures (MMC) for PHA production using residuals as feedstock are considered as the most promising technology to develop more inexpensive methods to produce PHAs, also extending their biocircularity [9,10,11,12]. In the second case, the incorporation of biowastes or agro-based byproducts such as vegetable fibers as fillers have been widely studied and appears to be an effective strategy to reduce costs while maintaining or even improving the properties of PHAs [17,18,19]. Regarding the limitations in mechanical behavior, reactive blending with elastomeric thermoplastics seems to be an efficient approach to enhance the toughness of brittle polymers by rubber toughening, where the dispersed elastomeric secondary phase acts as an impact modifier, increasing the impact-absorbed energy of the resin [20,21,22]. Reactive blending could also play an important role in improving the thermal stability if the compositions are properly formulated [23,24,25].

Rubber toughening has been studied in depth since the middle of the last century due to its importance at an industrial level [26,27,28,29,30]. It has been known for a long time and is well accepted that toughening depends on both the average size of the dispersed phase, its distribution, its concentration, and the interfacial adhesion with the polymeric matrix. Nevertheless, the specific mechanisms of rubber toughening have not yet been clarified and the determining factors in the toughening effectiveness continue to be a matter of study and controversy. In this regard, several theories have been proposed in the literature, with the void toughening theory being one of the most widely accepted. In this theory, which has been built over the years, it is postulated that when a mechanical stress is applied, the stress concentrates around the rubber particles, giving rise to their cavitation, followed by shear yielding and/or crazing thorough the polymeric matrix, thus absorbing much of the energy of the fracture [31].

The rubber toughening approach, for instance, has been applied to PHAs and other linear biopolyesters, exploring their blends with thermoplastic polyurethane (TPU) [32,33,34], biodegradable polymers such as poly(butylene-succinate) (PBS) [35,36], poly(butylene adipate-co-terephthalate) (PBAT) [37], and polycaprolactone (PCL) [38] or natural rubbers [39,40], among others. Indeed, the addition of TPU to PHBV has been studied in previous works of the group [32,41,42]. In these works, ternary PHBV/TPU/cellulose blends with enhanced toughness with respect to neat PHBV were successfully obtained by reactive extrusion, showing that hexamethylene diisocyanate (HMDI) had the highest ability at compatibilizing the phases among the different reactive agents tested [42]. Despite the good results obtained, more research is needed to better understand the interaction between the polymeric phases PHBV/TPU and the role that the HMDI plays at compatibilizing them and enhancing the toughness.

On the other hand, from the point of view of industrial applicability, it is important to deepen the study of the performance of this type of system in service as structural and or semi-structural applications since little information in this regard can be found in the literature. Therefore, it is particularly interesting to analyze the performance as a function of temperature, thinking of applications in which the material may be subject, for example, to seasonal temperature changes (e.g., in automotive parts), freezing, or heating (e.g., in food packaging applications). As well as long-term behavior, thinking, for example, of applications in which it will be subjected to constant loads (e.g., structural) [43,44,45].

Bearing this in mind, the objective of this work was the development and characterization of toughness-improved blends of PHBV/TPU using HMDI as a reactive extrusion agent. This study will provide key-information to set up the basis for understanding the role of the elastomer and compatibilizer on the rheological behavior, thermal properties, and long-term mechanical properties of the PHBV blends with the impact modifiers. Hence, in the roadmap to achieve a full biobased biodegradable formulation that fulfils all requirements in terms of technical performance, this knowledge will help to select ad-hoc bioelastomers and reactive compatibilizing agents. Since biodegradation is one of the main interests of these materials, biodegradability tests of the resultant blends were also investigated in the present work.

## 2. Materials and Methods

### 2.1. Materials

Poly(3-hydroxybutyrate-co-3-hidroxivalerate) (PHBV) commercial grade with 3 wt%. hydroxyvalerate content was purchased from NaturePlast (Ifs, France) in pellet form (PHI002). Thermoplastic polyurethane (TPU) Elastollan^®^ 890 A 10FC was kindly supplied by BASF (Ludwigshafen, Germany). The reactive agent used, hexamethylene diisocyanate (HMDI) was supplied by Sigma Aldrich (Madrid, Spain).

### 2.2. Sample Preparation

The PHBV and TPU used in this study were dried at 80 °C for 8 h in a DESTA DS06 HT dehumidifying dryer, while the reactive agent (HMDI) was used as received.

Neat PHBV and PHBV/TPU blends with and without the reactive agent (HMDI) were prepared in a Labtech LTE (Samutprakarn, Thailand) (Ø = 26 mm, L/D ratio = 40) with a three kneading blocks and a twin screw extruder. The temperature profile used was 140/150/175/175 °C from the hopper to nozzle and the rotation speed was 140 rpm. PHBV and TPU were manually premixed before extrusion (dry-blend) and fed to the main hopper by the extruder feeder. The HMDI was dispensed at the feeding zone (feed rate 32 rpm) by means of a peristaltic pump (Watson Marlow 120 S/R, Sondika, Spain). The extruded material was cooled in a water bath and pelletized.

The obtained pellets were dried again at 80 °C for 9 h before injection molding process. The samples were injection-molded in a DEMAG IntElect 100 T/470-340 machine (Schwaig, Germany) with an injection temperature of 185 °C at the nozzle and a cavity wall temperature of 65 °C. The injection time was of 0.5 sg, and then a holding pressure of 600 bar was applied for 12 s, followed by 40 s of cooling time.

The nomenclature used for naming the blends is as follows: PHBV, PHBV/TPU, and PHB/TPU/HMDI. Table 1 summarizes the samples studied and their compositions.

### 2.3. Morphology Characterization

The morphology of the PHBV, PHBV/TPU, and PHBV/TPU/HMDI blends were examined by scanning electron microscopy (SEM) using a high-resolution field-emission JEOL 7001F microscope (TokyoJapan). The fracture surfaces from the impact-fractured specimens were covered by sputtering with a thin layer of Pt prior to the SEM observations. From select representative SEM images (at 5000× magnification), the diameter of the droplets corresponding to the dispersed phase was measured using Fiji^®^ software (ImageJ 1.51j8) (National Institutes of Health, Bethesda, MD, USA). The number of droplets measured in all cases was higher than 600. From each blend, the following parameters were determined: the average droplet size in number (*d*) and the droplet size distribution parameters d10, d50, and d90 (corresponding to the size where 10%, 50%, and 90% of the droplets were included, respectively). The matrix ligament thickness (*T*) was also estimated, according to Wu’s Equation (1) [26]:(1)$$T=d\left[{\left(\frac{\pi }{6\varphi_{r}}\right)}^{1/3}-1\right]$$
where *d* is the average domain size of the dispersed phase and φ*_r_* is the volume fraction of the dispersed phase, determined as follows (2):(2)$$\varphi_{r}=\frac{\rho_{m}w_{r}}{\left(\rho_{r}w_{m}+\rho_{m}w_{r}\right)\prime }$$
where $\rho_{m}$ and $\rho_{r}$ are the densities of the matrix and dispersed phase, respectively, and $w_{m}$ and $w_{r}$ are their weight fraction.

### 2.4. Thermal Characterization

The thermal degradability of the blends was investigated by means of TGA (Mettler Toledo, Barcelona, Spain) using a TG-STDA Mettler Toledo model TGA/SDTA851e/LF/1600. The samples were heated from 30 to 900 °C at a heating rate of 5 °C/min under nitrogen flow. The characteristic temperatures, *T*_5%_ and *T_d_*, corresponded, respectively, to the initial decomposition temperature (5% weight loss) and to the maximum degradation rate temperature measured at the derivative thermogravimetric analysis (DTG) peak maximum.

Differential scanning calorimetry (DSC) (Mettler Toledo, Barcelona, Spain) experiments were conducted on a DSC (Mettler Toledo) with an intracooler (Julabo FT900) calibrated with an Indium standard before use. The weight of the DSC samples was between 4 and 6 mg. Samples were first heated from −40 °C to 190 °C at 10 °C/min and kept for 3 min at 190 °C, followed by cooling to −40 °C at 10 °C/min, and kept for 3 min, before finally heating to 190 °C at 10 °C/min. Melting temperatures *T_m_* and enthalpies (ΔH_m_) as well as the crystallization temperatures *T_c_* and enthalpies (ΔH_c_), were calculated from the second heating and cooling curves, respectively. In order to assess the influence of the secondary phase on the PHBV ability to crystallize, PHBV crystallinity (χ_c_) was calculated using a theoretical value of 146 J/g [46] for pure PHBV.

Dynamic mechanical analysis (DMA) experiments were conducted on injected samples (55 × 10 × 4 mm^3^) in an AR G2 oscillatory rheometer (TA Instruments, New Castle, DE, USA) equipped with a clamp system for solid samples (torsion mode). Samples were heated from −20 °C to a melting temperature with a heating rate of 2 °C/min at a constant frequency of 1 Hz. The maximum strain was set at 0.1%.

### 2.5. Rheology

Oscillatory shear measurements were performed with an AR G2 oscillatory rheometer (TA Instruments, New Castle, DE, USA) equipped with 25 mm parallel plates with a gap of 1.5 mm. Pellet samples were vacuum-dried at 80 °C overnight before testing in a Piovan DPA 10 (Santa Maria di Salva VE, Italy). Strain sweep viscoelastic tests were first performed at a fixed angular frequency of 1 Hz to determine the extent of the linear regime; then, the frequency sweep experiments were carried out at a fixed strain in the linear regime to determine the linear viscoelastic modulus, storage modulus (G′), loss modulus (G″), and complex viscosity (η). The angular frequencies were swept from 100 to 0.01 Hz at a temperature of 180 °C.

In order to evaluate the thermal stability of the blends, complex viscosity was monitored as a function of time. Time sweep measurements were conducted at 180 °C, and a fixed angular frequency of 1 Hz, with a gap of 1.5 mm for 900 s.

On the other hand, capillary rheometry characterization was also performed using a Rosand RH 7.2 Extrusion Capillary Rheometer (ASTM D5099). Measurements were made at 180 °C. Pellet samples were dried at 80 °C/8 h in a dehumidifier (DESTA DS06 HT), and the sample was preheated at test temperature for 8 min before the test started. A capillary of 0.5 mm diameter and 16 mm length was used; Rabinowitch was neglected. Viscosity data as a function of the apparent rate were measured.

According to the literature, in the non-Newtonian range, the relationship between apparent melt viscosity (η) and the shear rate $(\dot{\gamma }$) can be fitted using Equation (3) in the double logarithmic scale according to the Ostwald–deWaele Power law model [47]:(3)$$\eta =K{\left(\dot{\gamma }\right)}^{n-1}$$
where η is the apparent melt viscosity; *K* is the consistency parameter (temperature-dependent viscosity-like property); $\dot{\gamma }$ is the shear rate; and *n* is the flow behavior index. The value of η allows us to classify the type of fluid in Newtonian (*n* = 1), shear-thinning non-Newtonian (0 < *n* < 1), or shear thickening fluid (*n* > 1).

### 2.6. Mechanical Characterization

The mechanical characterization of the neat PHBV, PHBV/TPU, and the PHBV/TPU/HMDI blends was performed by tensile and flexural tests in a MTS Insight universal testing machine equipped with a 500 N and 100 KN load cell, respectively. Tests up to failure were conducted on type 1A injection-molded specimens according to the ISO-527 standard for tensile tests and ISO-178 standard for flexural tests. All of the studied materials were analyzed at three different temperatures: −20 °C, 23 °C, and 50 °C. For both tests, five samples were used for each temperature of measurement.

Additionally, the tensile creep tests of the samples were performed with MTS Insight under a stress value of 15 MPa at three different temperatures (−20, 23, and 50 °C). Tests were run according to ISO 899-1. Once the test device was thermally stabilized, the 15 MPa load was applied smoothly (load was applied in a time range of 1 to 5 sg) to the tensile specimen, and then kept constant. The resultant elongation of the sample was measured as a function of time, up to 200,000 s (55 h).

### 2.7. Biodegradation

Biodegradation tests in industrial composting conditions of PHBV, PHBV/TPU, and PHBV/TPU/HMDI were carried out according to the ISO 14855-1 standard. The bioreactors used were 2 L hermetic bottles. According to the standard, three replicates of each sample were made as well as a reference material (TC40 cellulose, from CreaFill Fibers Corp., Chestertown, MD, USA) and a blank. In each reactor, the ground powder sample was mixed with mature compost (previously sieved) and adjusted to 50% water content. The bioreactors were incubated at 58 °C for 90 days. Periodic measurements of the generated carbon dioxide and oxygen were taken to track the mineralization of the sample while ensuring aerobic conditions. The degree of biodegradation (B%) was calculated by comparing the accumulated CO_2_ at a given time of incubation with the theoretical total carbon dioxide using Equation (4):B(%) = [(CO_2_ (t) − CO_2_ (b))/ThCO_2_ ]·100(4)
where CO_2_ (t) is the accumulated carbon dioxide of the sample at a specific time; CO_2_ (b) is the average accumulated carbon dioxide of the blank at the same time; and ThCO_2_ represents the total theoretical carbon dioxide calculated from the total organic carbon and the mass of each sample.

## 3. Results

### 3.1. Morphology Characterization

The morphology of PHBV, PHBV/TPU, and PHBV/TPU/HMDI was analyzed by SEM. Figure 1a,b shows the micrographs of the pristine blend at different magnifications, whereas Figure 1c,d shows the corresponding micrographs of the blends compatibilized with HMDI.

As shown in Figure 1a,c, both blends presented a typical “drop in matrix” two-phase morphology, which is typically found in immiscible systems. This microstructure is characterized by TPU domains or drops homogeneously dispersed within the PHBV continuous matrix. The size distribution and the characteristic parameters of the dispersed TPU particles in the blends were obtained from the SEM micrographs. The results are summarized in Figure 2

The average size of the dispersed phase (TPU) in the PHBV/TPU blends was 0.46 µm and the estimated ligament distance was 0.114 µm. With the addition of HMDI, it can be observed that the average domain size as well as the ligament distance were significantly reduced to 0.34 µm and 0.104 µm, respectively.

Regarding the drop size distributions obtained (d10, d50, d90), HMDI was particularly efficient at reducing the presence of larger domains, as can be deduced from the d90 values that went from 0.9 in the PHBV/TPU blend to 0.65 in the PHBV/TPU/HMDI blend. In fact, the number of the biggest droplets, larger than 1.2 µm, was drastically reduced with the addition of HDMI.

The reduction in the size distribution of the TPU drops could indicate an increase in the interaction between the phases. According to this, some detachment of the TPU droplets could be observed in the blend without HMDI, as seen in Figure 1a,b. However, as shown in Figure 1d, with the addition of HMDI, no detachment was observed and even drops of TPU broken in half could be seen, suggesting increased adhesion between the phases [48,49,50,51].

According to the hypothesis of Wu [26], the fundamental factor to reach the maximum toughness is the inter-particle distance among the dispersed rubber droplets (matrix ligament thickness). From his conclusions, shear yielding occurred only below a critical ligament thickness so that plastic deformation percolated across the specimen. In contrast, Bucknall and Paul completely discarded this idea and concluded that the maximum toughness depends on the particle size and volume fraction of the rubber phase [27,28]. Very small particles are ineffective because they are more resistant to cavitation whilst large cavitated rubber particles act as craze-initiating Griffith flaws. Even in blends with quite small average particle sizes, a small percentage of larger particles would be sufficient to induce premature fracture, especially, if these are detached from the matrix. Thus, there is an optimal intermediate particle size range where the maximum toughness is achieved because the particles are large enough for complete cavitation preceding yielding. This range is typically comprised between 0.2 and 0.4 µm for most systems [27,28].

Hence, according to the morphological analysis, two key properties should be enhanced by the addition of HMDI to the blends:Increase in toughening, as the fraction of particles between 0.2 and 0.4 µm increases and the number of big particles (*d* > 0.6 µm) decreases.Increase in melt viscosity due to the increased interfacial adhesion between PHBV and TPU.

### 3.2. Thermal Characterization

#### 3.2.1. Thermogravimetric Analysis

TGA was conducted on the raw materials and the blends to evaluate the thermal degradation of the PHBV and PHBV/TPU blends with and without HMDI. The resulting TG and DTG curves are depicted in Figure 3. From these curves, the onset degradation temperature (*T*_5%_, considered as the temperature at which a 5% weight loss occurs) and the maximum degradation temperature (*T_d_*), corresponding to the DTG peak, were calculated and their values are summarized in Table 2.

As it is well reported, PHBV thermal degradation takes places abruptly in a single weight loss step with an onset at 278 °C and the maximum degradation temperature at 298 °C [52]. The PHBV/TPU blends showed two degradation stages, of which the first stage was ascribed to the PHBV degradation while the second stage was a result of the TPU degradation (Figure 3). The onset degradation and the maximum degradation temperatures of the PHBV/TPU blends with and without HMDI were comparable to those corresponding to neat PHBV. However, with the addition of TPU and HMDI, a slight change in the slope of the mass loss curve was observed in the first stage of degradation.

#### 3.2.2. Differential Scanning Calorimetry

The thermal behavior of the PHBV, PHBV/TPU, and PHBV/TPU/HMDI blends were also analyzed by DSC. The main thermal parameters, obtained from the second heating scans after thermal history erasing, are summarized in Table 3.

Regarding the results obtained, the addition of TPU to PHBV produced a slight decrease in the crystallization temperatures of the blends studied, indicating that this component hinders the crystallization process. Such behavior can be explained by the intermolecular interactions between the phases in the liquid state, introducing some disorder in the system, which would hamper the crystallization. This is consistent with the slight decrease in the χ_c_ and melting temperature, indicating that the presence of the second phase would result in a less perfect crystalline structure. No remarkable differences were observed with the incorporation of HMDI to the PHBV/TPU blends [32,42]. This indicates that the reactivity of HMDI did not take place at random places of the polymer chains, as expected.

#### 3.2.3. Dynamic Mechanical Analysis

DMA was performed on the PHBV, PHBV/TPU, and PHBV/TPU/HMDI blends to study the effect of TPU (with and without HMDI) on the mechanical performance throughout the whole temperature range from the glass transition temperature (*T_g_*) to melting temperature. The storage modulus (G′) and Tan-δ curves are represented in Figure 4.

With respect to the storage modulus (G′), the PHBV showed a relatively high stiffness in the entire temperature range studied. By incorporating the elastomer, a clear reduction in stiffness was observed. Indeed, the greater height of the damping factor (tan δ) indicates a greater contribution of the viscous component of the complex modulus (G″). Tan δ peaks are usually employed to determine the glass transition temperature (*T_g_*) in semicrystalline polymers [53]. Two events/peaks were observed in the blends with TPU, the first at around −14 °C, corresponding to the *T_g_* of TPU, and the second at about 28 °C, corresponding to PHBV *T_g_*. No remarkable differences were observed in the tan δ peak position with the addition of HMDI, thus confirming that no random interaction of the compatibilizing agent affects the amorphous fraction in the polymer matrix. Therefore, the interaction of HMDI with the polymers is restricted to their interphase.

### 3.3. Rheology

#### 3.3.1. Plate-Plate Rheology

Dynamic oscillatory rheology is an effective tool to investigate the polymer–polymer and polymer–compatibilizer interactions in the molten state of immiscible polymer blends. This is of particular interest when it comes to studying the interaction of phases in a heterogenous blend. To evaluate the influence of TPU and HMDI in the rheological behavior of PHBV, frequency sweep tests were carried out in the linear viscoelastic range. The evolution of the storage modulus (G′), loss modulus (G″), and complex viscosity (η*) as a function of frequency are shown in Figure 5.

For neat PHBV, G′ and G″ increased with frequency, showing a more elastic (solid-like) response at high frequencies and more viscous response at low frequencies. This behavior can be explained by the viscoelastic (time dependent) nature of the polymer; at high frequencies, the polymer chains do not have enough time to relax and the elastic response of the melt predominates, whilst the opposite occurs at low frequencies [54]. When TPU was added, higher values of G′ and G″ were obtained over the entire frequency range analyzed with respect to TPUa due to the highest elasticity and viscosity of this component. In addition, a small shoulder in G′ could be observed below frequencies of 1 Hz, which is characteristic of immiscible blends with a drop in matrix morphology and can be related to the relaxation of the droplets of the dispersed phase [55]. When HMDI was added, significant changes in the viscoelastic response of the blend were observed in the terminal region (low frequency range). In this case, a clear increase in G′ showed a pseudo-solid-like behavior at this frequency region. This trend is usually observed in nanoparticle filled polymers and blends due to the particle–particle interconnectivity of nanoparticles and the formation of a pseudo-network structure [56,57]. However, it is not easy to find a simple explanation for this trend in the unfilled blends. This is attributed in the literature to the formation of complex morphologies showing slow relaxation dynamics [55]. These observations could be related to the enhanced adhesion between the polymeric phases induced by the HMDI, thus resulting in an increase in the droplet–matrix surface, leading to an increased elasticity and viscosity of the melt [48,57].

With respect to complex viscosity, the PHBV and PHBV/TPU blends presented a Newtonian plateau followed by a decrease in viscosity at high frequencies. On the other hand, the slight decrease in viscosity observed at low frequencies in these compositions can be attributed to thermal degradation after the long thermal exposure during the experiment [58]. The incorporation of HMDI produced an increase in the viscosity and shear thinning behavior across the entire range of frequencies. The same trend observed in G′ was detected in complex viscosity for this blend. Again, the improved melt viscosity of the blend can be explained by the enhanced adhesion of the phases due to the compatibilization effect of HMDI. These results are in agreement with the SEM observations, where better attachment of the polymer matrix on the TPU entities was revealed.

The influence of TPU and HMDI on the thermal stability of neat PHBV was also examined by the rheological tests. PHBV as well as other biopolyesters experiences fast degradation during processing. In the particular case of PHBV, the mechanism underlying the fast thermal degradation of PHA is the hydrolysis of the ester group, thus leading to a reduction in molecular weight [59]. In addition, due to is natural origin, the presence of chemical impurities also contributes to worsening this problem [60]. Dynamic oscillatory time sweep tests at 180 °C were conducted in order to elucidate whether TPU and/or HMDI addition could help to mitigate this shortcoming. The evolution of complex viscosity normalized at its initial value as a function of time is depicted in Figure 6.

As can be deduced from the plots in Figure 6, a considerable reduction in the viscosity drop was observed in the PHBV/TPU blends with respect to the pure PHBV. The addition of HMDI to the blend did not alter this behavior with regard to the normalized viscosity, even though the absolute viscosity values were much higher in the compatibilized blend (as shown in Figure 5). While the neat PHBV and PHBV/TPU blend showed comparable initial viscosities of around 500 Pa·s, the compatibilized blend showed an initial viscosity of about 1200 Pa·s. Hence, together with the results obtained from the TGA, this led us to conclude that the compatibilized blends could improve their processing stability with respect to the neat PHBV or uncompatibilized blends.

#### 3.3.2. Capillary Rheometry

The shear viscosity behavior of the neat PHBV and the PHBV/TPU blends was studied by capillary rheometry in the range of injection shear rates. The obtained results are represented in Figure 7. The rheological parameters, K and n, obtained with the Ostwald–deWaele Power law model [47], and the injection pressures are summarized in Table 4.

Regarding the apparent viscosity curves in Figure 7, all of the compositions presented a non-Newtonian behavior with shear thinning characteristics (0 < *n* < 1). The addition of TPU to PHBV led to an increase in the apparent viscosity across the whole range of shear rates tested, which was enhanced by the presence of HMDI. This indicates the reactivity in the blend. Nevertheless, no remarkable differences in the pseudoplastic behavior were observed, as the shear thinning index remained practically unchanged. When comparing the consistency coefficients (which are viscosity dependent) with the registered injection pressures, similar trends were observed, with higher values for both parameters as TPU and TPU/HMDI were added. These results are in agreement with the dynamic rheological results and the performance of the blends during processing, also confirming the enhanced interaction between the polymeric phases due to HMDI compatibilization.

### 3.4. Mechanical Characterization

To deeply understand the influence of TPU on the mechanical performance of PHBV and the role of HMDI as the compatibilizer, the mechanical behavior of the compositions was analyzed by uniaxial tests up to break, flexural tests, tensile creep tests, and dart drop impact tests. The influence of the temperature on the mechanical performance and long-term behavior were analyzed.

#### 3.4.1. Tensile Test

The tensile modulus of elasticity (E), tensile strength (σ_max_), and elongation at break (ε_r_) of the blends and neat PHBV at the three temperatures tested (−20, 23 and, 50 °C) are represented in Figure 8.

As expected, PHBV presented a brittle and stiff behavior, with low deformation at break. This behavior is logically more marked at a lower temperature, especially below the glass transition. The addition of the elastomeric phase, TPU, results in a reduction in the rigidity of the samples (E), and the tensile strength (σ_max_), but an increase in the elongation at break (ε_r_). The positive effect of TPU at enhancing ductility can be related to the homogeneous and well-dispersed secondary phase, the small droplet size, and ligament thickness, as was observed by SEM. When HMDI was added, a significant additional improvement in the elongation at break was observed, even at the lowest temperature (−20 °C), that is, below the glass transition of both polymers. In fact, at this temperature, an elongation at break superior to that corresponding to the neat PHBV at room temperature was achieved and comparable to that corresponding to 50 °C. In addition to these improvements in ductility, the tensile strength was also enhanced with respect to PHBV/TPU without the reactive agent, maintaining a similar rigidity and indicating a clear improvement in the static toughness.

These results are in agreement with an effective compatibilization effect of HMDI, which led to the enhanced interfacial adhesion between the polymeric phases. This was already stated by the SEM observations, where the morphological variations allowed us to predict the toughness enhancement of the compatibilized blends. The increase in the toughening effectiveness of the TPU droplets regarding their size distribution allowed for more plastic deformation. In addition, the increase in the interfacial adhesion between PHBV and TPU (as evidenced in the rheological measurements) added some deformation ability to the soft domains before cavitation took place at the matrix–particle interface.

#### 3.4.2. Flexural Tests

In Figure 9, the bending modulus (E) and peak strength (σ_max_) of the PHBV and PHBV/TPU blends with and without HMDI are represented.

Regarding the flexural modulus (Figure 9), a similar trend as the one found in tensile configuration was observed. The incorporation of the elastomeric phase resulted in a decrease in the bending modulus and peak strength. The addition of HMDI was hindered by the test geometry, since the tensile and compressive stresses were combined and only the surface was subjected to the maximum forces. Nevertheless, slightly higher values for the flexural mechanical parameters of the compatibilized blends can be predicted.

#### 3.4.3. Tensile Creep

Tensile creep can be defined as the time-dependent deformation of a material under a constant uniaxial tensile load. This is of particular interest from an industrial point of view, in which to study the creep at different temperatures in order to predict the long-term behavior of the materials in service [61]. The typical creep response is characterized by an instantaneous elastic response (instantaneous strain), followed by the primary creep region, where the response is dominated by elastic–plastic deformation. The strain rate (slope in the strain vs. time graph) decreased rapidly in the primary creep region until it reached a stationary value, leading to the secondary creep region. This was characterized as showing a constant slope where the viscous–viscoelastic flow dominated, and the duration was very long with respect to the primary creep region. Finally, the tertiary creep region was characterized by an increase in the strain rate and final failure [61]. The creep curves of the PHBV and PHBV/TPU blends at −20, 23, and 50 °C are represented in Figure 10, where the primary and secondary regions can be appreciated.

At −20 °C, apart from the lower instantaneous deformation for the neat PHBV as a result of the more rigid elastic behavior, small differences among the different samples were observed in the creep response, presenting high creep resistance as deduced from the constant strain observed in the primary and secondary creep regions.

At room temperature, the addition of the TPU led to a higher instantaneous elastic deformation, in agreement with the lower tensile modulus found previously in the tensile tests. When looking at the curves in the secondary creep region, it can be noticed that the slopes of the neat PHBV and PHBV/TPU with HMDI were similar, whilst the slope of the uncompatibilized blend was higher. This response indicates a better creep performance of the blend with HMDI, similar to that of PHBV. Hence, with the incorporation of HMDI, although the instantaneous elastic response was similar to the PHBV/TPU blend, the creep resistance in terms of the strain rate was similar to that of the neat PHBV. At the highest temperature (50 °C), a similar trend could be discerned, although the differences among the samples were more pronounced.

This behavior can be explained, again, by the effective increase in the PHBV/TPU interfacial adhesion promoted by the reactive agent. Under a constant load, if there is poor adhesion at the interface, there is a time when the dispersed droplets can be detached, and the response becomes similar to that of a matrix with small holes or voids (with a consequent progressive reduction in the effective cross section of the sample). In contrast, with enhanced interfacial adhesion, the long-term response was mainly dominated by the matrix. Furthermore, the smaller size of the TPU droplets may also play a role in increasing the creep resistance, since small particles are more efficient at hindering the polymer structure evolution than the large sized ones [62]. These results also confirm the performance enhancement by increasing the compatibility of the PHBV/TPU blend with HMDI.

#### 3.4.4. Dart Drop Impact Tests

Dart drop tests were conducted to analyze the puncture impact behavior, closely related to the impact toughness of the PHBV and the PHBV/TPU blends with and without HMDI. The tests were carried out at −20, 23, and 50 °C. The force required to punch out the samples (which is proportional to impact energy) is represented in Figure 11.

The results show that the neat PHBV presented a very low F_max_ in dart impact, regardless of the temperature. This can be explained on the basis of its high crystallinity, which dominates the response at high deformation rates, with a brittle behavior. Surprisingly, at −20 °C, despite the improvements in the deformation at break of the PHBV/TPU blends reported in the tensile tests, their impact response did not show any increase in the peak value. According to the void rubber toughening theory, the stress concentration, deformation, and cavitation of the dispersed droplets followed by shear yielding/or crazing of the matrix is needed for energy absorption and thus toughening [63]. In this case, the conditions are very demanding for an intrinsic brittle matrix since the low temperature combined with the high speed does not allow for the necessary chain mobility required for toughening to occur, thus leading to a natural heterogeneous system where any failure in the interface between the PHBV and TPU droplets may act as critical flaws. [64]. According to ISO-6603, all samples can be classified as NY (non-yielding) and some of them, as indicated with an asterisk in Figure 11, as splintering materials.

On the other hand, at 23 and 50 °C, clear improvements in the impact behavior were observed when TPU was added. Moreover, a significant improvement was observed when HMDI was added. In fact, the F_max_ needed to punch out the sample practically doubled. As mentioned, different factors influence the effectiveness of rubber toughening, with it being accepted that interfacial adhesion, the size of the dispersed phase particle distribution as well as the distance between particles play an important role. However, there is some controversy about which of them are determinants to achieve maximum toughness.

Regardless of this controversy, in our case, HMDI proved its efficiency at reducing the average particle size of the dispersed phase, in particular, the size of the larger TPU domains, resulting in the reduction in the average ligament thickness and demonstrating a clear compatibilizing effect that results in significant toughness improvement.

### 3.5. Biodegradation

Figure 12 shows the evolution of biodegradation versus time of the neat PHBV and PHBV/TPU blends with and without HMDI. The biodegradation of these materials was compared with a refence cellulose sample (TC40). Polyurethanes are known to be susceptible to biodegradation [62]. Our initial hypothesis is that if the TPU is dispersed in very small droplets, the specific surface can be high enough to allow for a significant biodegradation percentage within the standard biodegradation testing. For this reason, biodegradation testing was attempted with the samples.

As shown in Figure 12, the biodegradation process occurred faster in the first 25 days. Later, the biodegradation rate decreased and tended to a plateau, where no additional increments in CO_2_ production were detected. All of the samples tested reached their maximum biodegradation at 55 days.

According to the results, PHBV can be considered as fully biodegraded in the tested conditions (after 60 days, reaching a biodegradability > 90%). In contrast, the PHBV/TPU and PHBV/TPU/HMDI blends achieved a value of nearly 75% after 90 days. Therefore, the blends cannot be classified as biodegradable according to the current standards (ISO 14855-1). Nevertheless, according to CO_2_ generation, at least 5% of the TPU was biodegraded during the period of testing, which indicates that perhaps over a longer period, these blends could be biodegraded. Bearing in mind the successful results obtained in improving the mechanical performance of PHBV, it would then be of great interest to explore the replacement of this TPU with some new generation bioelastomers that would not compromise biodegradability [65,66], or use special additives to promote the biodegradability of TPUs.

## 4. Conclusions

Toughness-improved PHBV/TPU blends were obtained by reactive extrusion by using HMDI as the reactive agent. HMDI demonstrated its efficiency as a compatibilizer by playing a key role in improving the overall performance of the blends. The PHBV/TPU blends presented a drop in the matrix morphology where TPU was the dispersed phase acting as an impact modifier. The incorporation of the HMDI led to a reduction in the overall particle size of the TPU droplets, in particular, the larger domains of TPU, and an enhanced adhesion of the polymer phases, as can be deduced from the SEM and rheological analyses. The increased compatibility between the phases led to a clear improvement in the ductility across the entire range of temperatures tested and an important improvement in the impact toughness at temperatures over the glass transition of the elastomer. The creep resistance of the PHBV/TPU blend was also improved by the incorporation of HMDI. In addition, reactive blending with TPU also helped to slightly minimize the fast thermal degradation of PHBV, as was observed in the TGA and rheological characterization. However, regarding the biodegradability, TPU compromised the full biodegradation of PHBV under standard conditions. Even though the blends are not considered biodegradable under the current standards, this work shows the path to improving the properties of PHBV to turn it into a plastic that can find applications in injected parts, solving one of the most critical limitations, which is its brittleness.

## Figures and Tables

**Figure 1 polymers-14-02337-f001:**
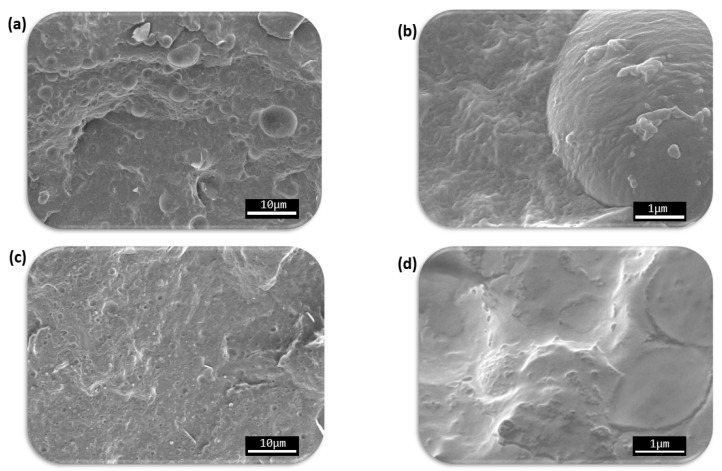
The SEM results. (**a**,**b**) SEM micrographs of the PHBV/TPU blend; (**c**,**d**) SEM micrographs of the PHBV/TPU/HMDI blend.

**Figure 2 polymers-14-02337-f002:**
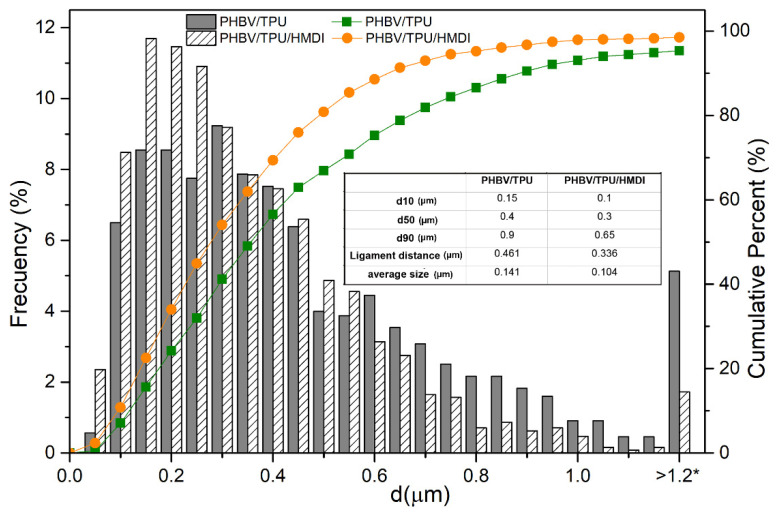
The particle size distributions of the TPU droplets in the blends, where the effect of the addition of HMDI is evidenced. * Frequency of droplets larger than 1.2 µm.

**Figure 3 polymers-14-02337-f003:**
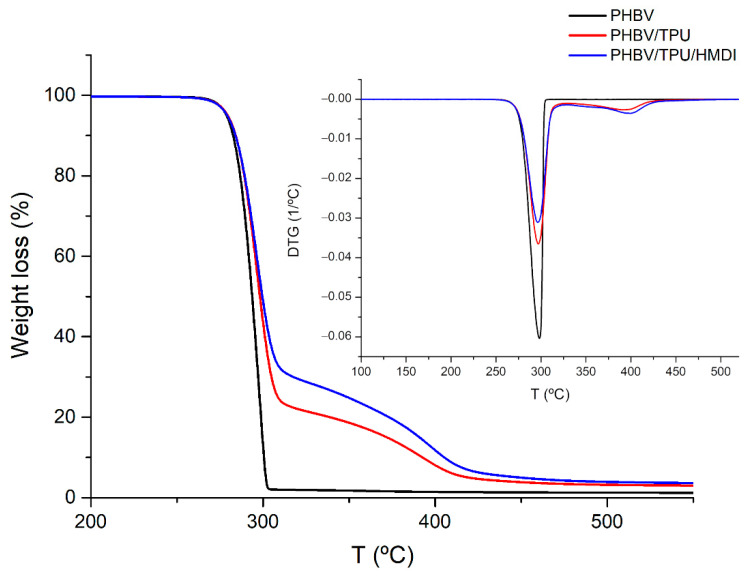
The TGA and DTG curves of the neat PHBV, PHBV/TPU and PHBV/TPU/HMDI blends.

**Figure 4 polymers-14-02337-f004:**
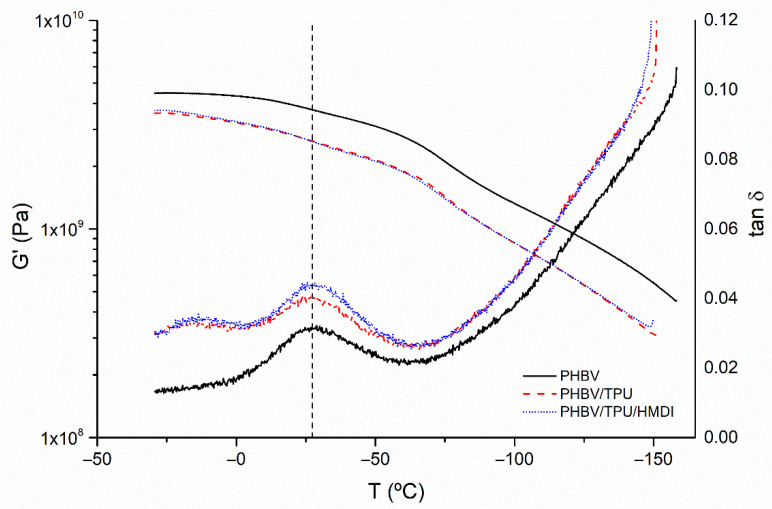
The storage modulus (G′) and tan δ evolution with temperature for the neat PHBV and PHB/TPU blends.

**Figure 5 polymers-14-02337-f005:**
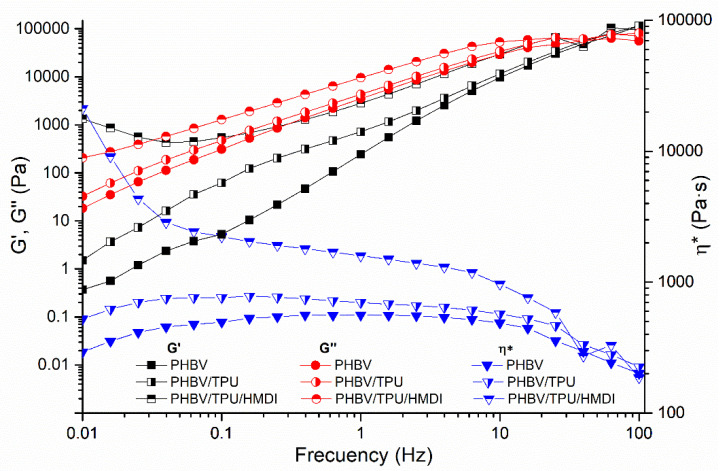
The storage modulus (G′), loss modulus (G″), and complex viscosity (ƞ*) as a function of the frequency of neat PHBV and PHBV/TPU blends.

**Figure 6 polymers-14-02337-f006:**
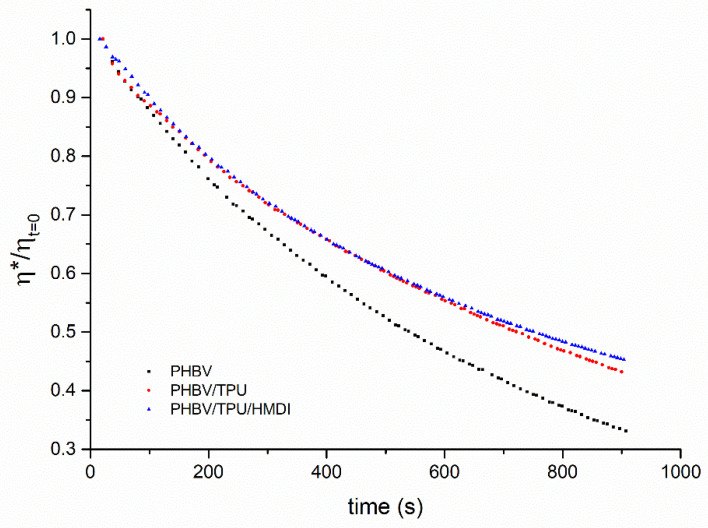
The normalized complex viscosity as a function of time of the neat PHBV and the PHBV/TPU blends.

**Figure 7 polymers-14-02337-f007:**
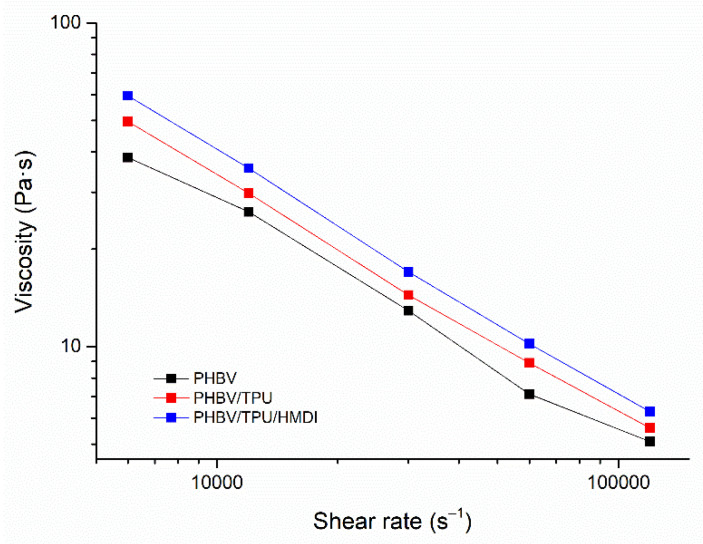
The viscosity as a function of thee apparent shear rate of the neat PHBV and the PHBV/TPU blends.

**Figure 8 polymers-14-02337-f008:**
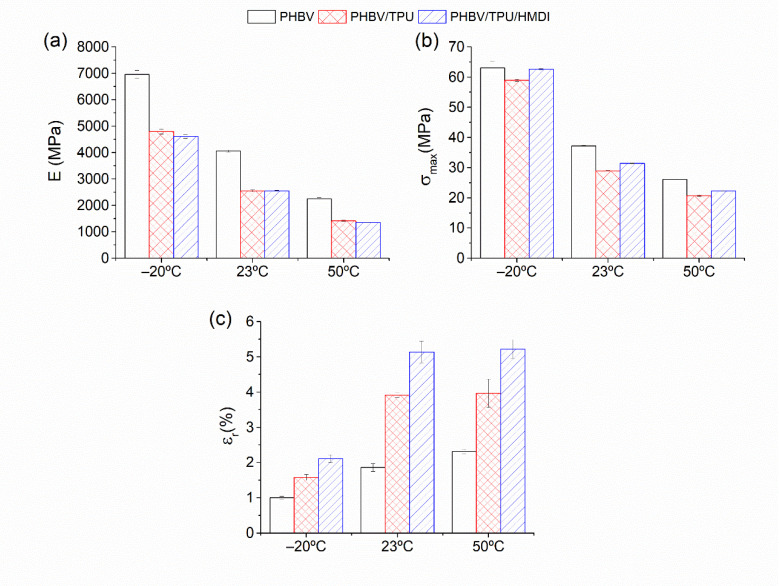
The tensile mechanical properties of the neat PHBV and the PHBV/TPU blends as a function of testing temperature: (**a**) Modulus of elasticity, (**b**) tensile strength, and (**c**) strain at break.

**Figure 9 polymers-14-02337-f009:**
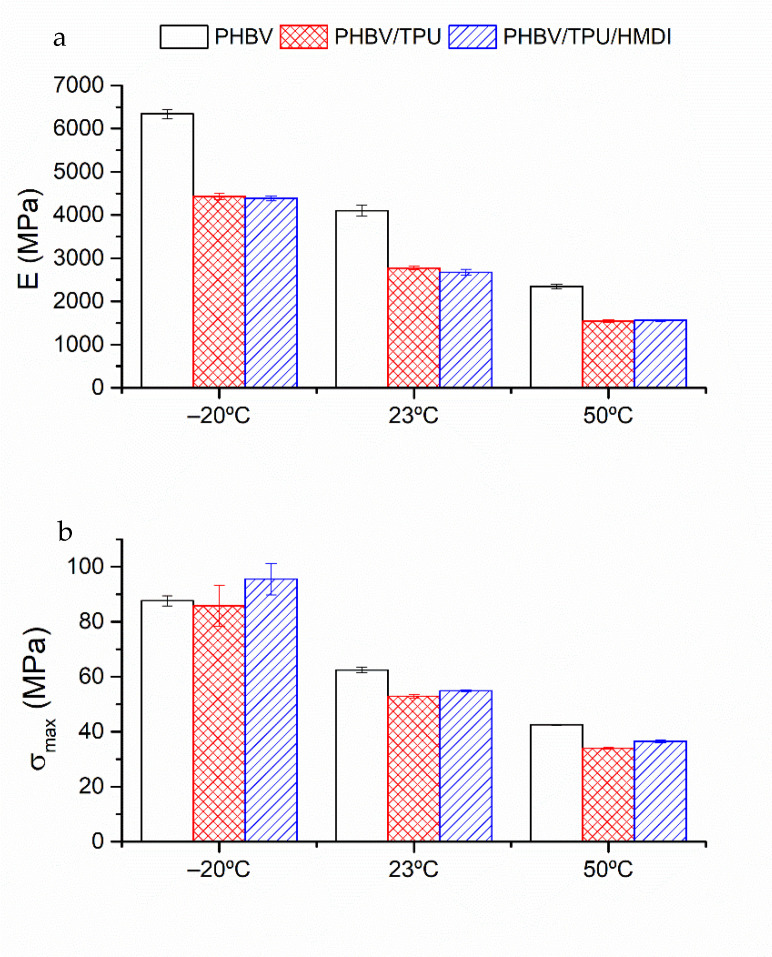
The flexural mechanical properties as a function of temperature of the neat PHBV and PHBV/TPU blends: (**a**) Modulus of elasticity, (**b**) Flexural strength.

**Figure 10 polymers-14-02337-f010:**
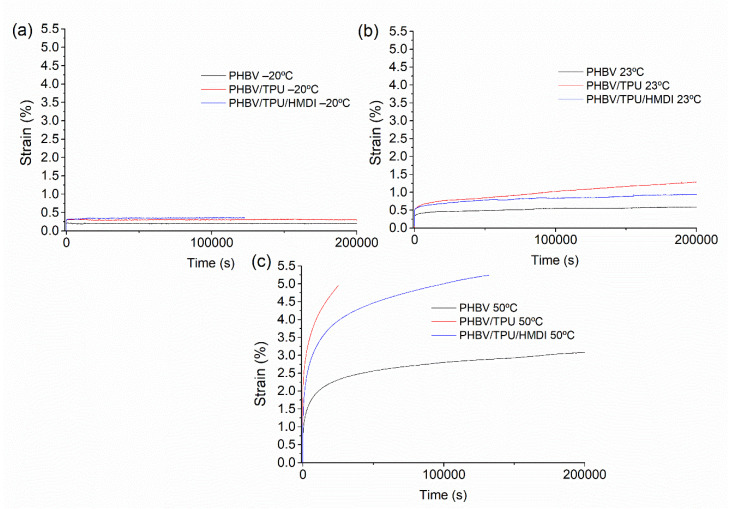
The tensile creep curves at 15 MPa of the neat PHBV and PHBV/TPU blends at (**a**) −20 °C, (**b**) 23 °C, and (**c**) 50 °C.

**Figure 11 polymers-14-02337-f011:**
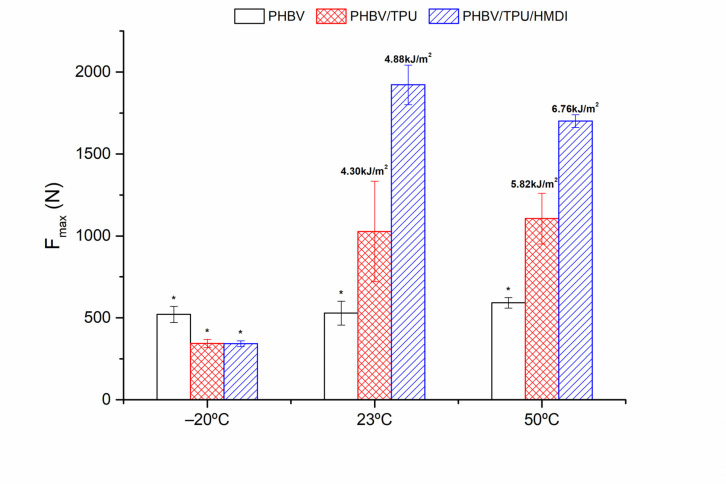
The maximum force in the dart drop impact test. The absorbed impact energy is indicated on top of each bar in the figure. Asterisks (*) indicate that these samples were broken with the formation of splinters.

**Figure 12 polymers-14-02337-f012:**
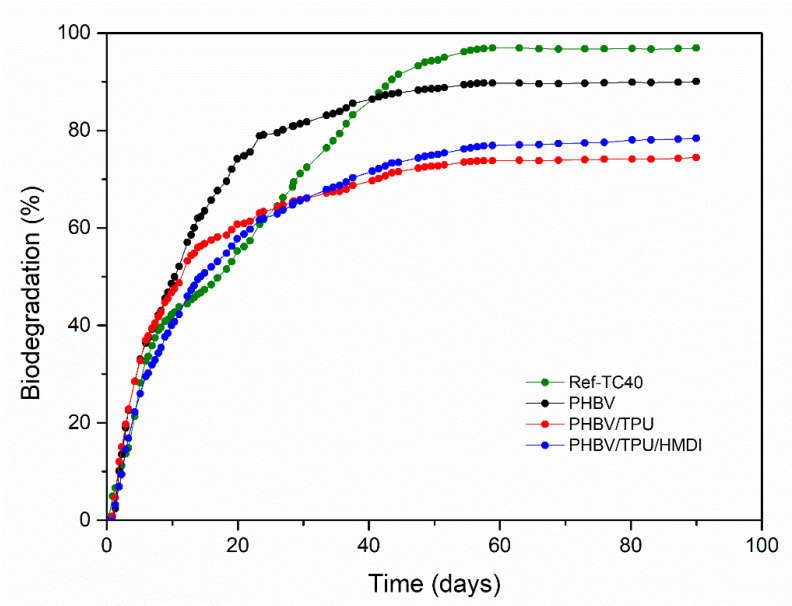
Biodegradation (%) in the composting conditions as a function of time.

**Table 1 polymers-14-02337-t001:** The nomenclature and composition of the samples.

Sample	PHBV (wt%)	TPU (phr)	HMDI (phr)
PHBV	100	0	0
PHBV/TPU	100	30	0
PHBV/TPU/HMDI	100	30	1

**Table 2 polymers-14-02337-t002:** The TGA parameters.

Samples	*T_d_* (°C)	*T*_5%_ (°C)
PHBV	298.2	278.7
PHBV/TPU	297.0	279.5
PHVB/TPU/HMDI	296.3	279.2

**Table 3 polymers-14-02337-t003:** The thermal properties according to the DSC curves.

Sample	*T_m_* (°C)	ΔH_m_ (J/g)	χ_c_ (%)	*T_c_* (°C)	ΔH_c_
PHBV	173.1	97.4	66.7	124.0	91.0
PHBV/TPU	171.4	72.2	64.3	113.0	65.6
PHVB/TPU/HMDI	171.5	70.7	63.4	112.3	68.7

**Table 4 polymers-14-02337-t004:** The processing injection pressure and capillary rheometry parameters.

Samples	Injection Pressure (bar)	Consistency Coefficient, *K*, (Pa·s*^n^*)	Flow-Behavior Index, *n* (−)
PHBV	450	18,160	0.296
PHBV/TPU	520	29,030	0.266
PHVB/TPU/HMDI	590	42,950	0.243

## Data Availability

Not applicable.

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
