# Peer review of "In Service Performance of Toughened PHBV/TPU Blends Obtained by Reactive Extrusion for Injected Parts"

_polymers, 2022, doi:10.3390/polym14122337_

Round 1
Reviewer 1 Report
Dear authors,
I have a note:
In figure 2 the distribution of the size of the droplets is shown. It does not clearly explain why the largest value (1.2) is greatly increased. Were all counted here, including those that are greater than 1.2. However, that would also mean that the calculated distributions are not correct.
In any case, a detailed explanation with a few sentences is necessary here. Please add this.
BR
The reviewer
Reviewer 2 Report
In the research, the toughness-improved blends of PHBV with thermoplastic polyurethane (TPU) have been developed and characterized.
The paper is valuable, and, in general, well written. However, the presented research is mainly of engineering value.
What does it mean: “Rabinowitch was neglected” ?
The material under study is strongly non-Newtonian (n<0.3). Why the Rabinowitsch correction was neglected. How about the Bagley correction ?
Author Response
Bagley's correction goal is to eliminate the pressure associated to the entrance effect into the capillary. When capillary measurements are run using a capillary with a L/D ratio over 25, it is assumed in literature that this pressure consumption in the entrance is much lower than the pressure consumption across the capillary flow, so it is neglected. In our case, capillary L/D was 32, so we neglected Bagley correction
About the Rabinowitch' one, it is a correction that allows to calculate the real shear rate and real viscosity data starting from apparent shear rate and viscosity data. We could run this correction, but our goal when running these capillary measurements was to get a comparison of the behaviour of the 3 different materials, and not getting the corrected o real data, so just for a comparison purpose it is enough to work with apparent shear rate and viscosity data.
Reviewer 3 Report
The study deals with the improvement of the properties of a biopolymer to be used in injected parts such as rigid packaging. In more detail, a poly(3-hydroxybutyrate-co-3-valerate) PHBV, was combined with an elastomer based on thermoplastic polyurethane (TPU) and a compatibilizer agent based on hexametylene diisocianate (HMDI) by a reactive extrusion process. Moreover, further characterization was employed addressing the morphological, rheological, thermal and mechanical properties of the produced blends resulting in a material with improved properties.
Generally, the work was well conducted; however, the authors publish a previous manuscript in the Molecular Sciences journal (2018) entitled “Toughness Enhancement of PHBV/TPU/Cellulose Compounds with Reactive Additives for Compostable Injected Parts in Industrial Applications”, that is the cited bibliographic reference 42. In this study PHBV/TPU/cellulose blends were produced by reactive extrusion using the same elastomer TPU and the same compatibilizer agent HDMI for the same purpose of been used in injected parts and to promote the toughness.
The author also indicates in the introduction “Despite the good results obtained, more research is needed to better understand the interaction between the polymeric phases PHBV/TPU and the role that plays the HMDI at compatibilizing them and enhancing the toughness.”. This reviewer also agrees with the sentence; however, no chemical characterization was considering in the experimental design to understand and address the chemical interactions in those blends and to support the findings. Due to the lack of novelty, this reviewer does not recommend the manuscript for publication in this journal.
Author Response
We kindly appreciate the reviewer’s point of view, addressing the lack of further chemical research on the role of the HMDI on the blend. However, we disagree in the fact that no further research was performed with respect to the previous paper, just because there are no direct chemical characterizations.
In this work, the toughness effect of the compatibilized blend has been thoroughly studied. This research has been approached in impact tests at different temperatures, analysis of particle size from SEM pictures and this data has been correlated with thermal and rheological properties. Indeed, the biodegradation analysis has proven that the potentially harmful isocyanates present in hexamethylene diisocyanate (HMDI) do not compromise the biodegradability of the PHBV fraction, hence they could most likely be used in other systems that are designed to be fully biodegradable.
Altogether, this paper offers novel data, not very often described in the literature, of PHBV impact modified blends and their behaviour under creep, impact tests, capillary rheometry and biodegradation performance under composting conditions. This may serve for other researchers as reference data for future works, meant to improve materials, which are to replace others for being more environmentally friendly under appropriate circumstances. And in all this results, the key has been the role of the HMDI, whose reaction between matrix and particles have produced significant improvements.
Round 2
Reviewer 3 Report
The author addresses the main questions raised by the reviewers, however the absence of a version highlighting the sentences or corrections performed are always very appreciated to identify the improvements. In general, this version was improved and it is more clear and discussed. Thus, the reviewer suggests to accept the manuscript in the current form for publication.